# Mitochondrial HSP70 Chaperone System—The Influence of Post-Translational Modifications and Involvement in Human Diseases

**DOI:** 10.3390/ijms22158077

**Published:** 2021-07-28

**Authors:** Henrieta Havalová, Gabriela Ondrovičová, Barbora Keresztesová, Jacob A. Bauer, Vladimír Pevala, Eva Kutejová, Nina Kunová

**Affiliations:** Institute of Molecular Biology, Slovak Academy of Sciences, Dúbravská cesta 21, 845 51 Bratislava, Slovakia; henrieta.havalova@savba.sk (H.H.); gabriela.ondrovicova@savba.sk (G.O.); barbora.keresztesova@savba.sk (B.K.); jacob.bauer@savba.sk (J.A.B.); vladimir.pevala@savba.sk (V.P.)

**Keywords:** mtHSP70, mortalin, HEP1, TID-1, GRPE, mitochondrial chaperones, post-translational modification, protein quality control, neurodegenerative disorders, cancer

## Abstract

Since their discovery, heat shock proteins (HSPs) have been identified in all domains of life, which demonstrates their importance and conserved functional role in maintaining protein homeostasis. Mitochondria possess several members of the major HSP sub-families that perform essential tasks for keeping the organelle in a fully functional and healthy state. In humans, the mitochondrial HSP70 chaperone system comprises a central molecular chaperone, mtHSP70 or mortalin (HSPA9), which is actively involved in stabilizing and importing nuclear gene products and in refolding mitochondrial precursor proteins, and three co-chaperones (HSP70-escort protein 1—HEP1, tumorous imaginal disc protein 1—TID-1, and *Gro-P* like protein E—GRPE), which regulate and accelerate its protein folding functions. In this review, we summarize the roles of mitochondrial molecular chaperones with particular focus on the human mtHsp70 and its co-chaperones, whose deregulated expression, mutations, and post-translational modifications are often considered to be the main cause of neurological disorders, genetic diseases, and malignant growth.

## 1. Introduction

The mitochondrial protein quality control (PQC) system is a network-like organization of chaperones and proteases whose major role is to preserve the functional and active states of mitochondrial proteins under diverse, and sometimes pathogenic, conditions. PQC maintains protein homeostasis or proteostasis among the mitochondrial proteins by controlling the balance between the generation of newly synthesized proteins and the removal of damaged or misfolded proteins beyond the scope of repair or refolding.

Protein damage, in the context of its structure and functionality, usually means a loss of function combined with changes to its native conformational state, both of which might occur as a result of non-physiological temperatures (heat stress) or chemical modifications (e.g., reactive oxygen species (ROS)) causing oxidative stress, which in humans may lead to a wide variety of pathologies, including cancer, neurodegenerative disorders, diabetes, cardiovascular diseases, atherosclerosis, stroke, inflammatory disorders, chronic fatigue syndrome, asthma, and age-related pathologies (review in [1]). Misfolded proteins are prone to form abnormal or irregular interactions as a result of the exposure of their normally buried hydrophobic parts; this often leads to the toxic accumulation of insoluble aggregates [2]. Interestingly, even a slight increase in temperature can trigger a heat shock response. The problem is not the temperature itself, but the protein unfolding, entanglement, and non-specific aggregation that the heat causes. In eukaryotes, the major damage appears to consist of disruption to the cytoskeleton (reorganization of actin filaments and tubulin networks), fragmentation of the membranous organelles (the Golgi complex and endoplasmic reticulum), aggregation of ribosomal proteins, and a decrease in the number of mitochondria and lysosomes [3]. Mitochondria loss leads to a dramatic drop in oxidative phosphorylation and ATP levels, causing a further imbalance in homeostasis, ultimately leading to cell death [4]. To avoid this fate, cells possess so-called heat shock proteins (HSPs), whose expression increases in response to such stresses, and which serve as the foundation of resistance to hostile conditions [3].

HSPs primarily act as molecular chaperones that monitor and regulate polypeptide folding and prevent the aggregation or precipitation of misfolded proteins. They are often overexpressed upon stress, in situations that would normally be lethal. Oxidative, cytokine and muscular stresses; nutritional deficiencies; viral infections; hyperthermia; ischemia and alterations in calcium and pH in different types of cells and tissues are all potent inducers of increased HSP levels. In humans, their deregulation often underlies the pathologies of several diseases, including such devastating neurological disorders as Alzheimer’s disease (AD), Parkinson’s disease (PD), Creutzfeldt-Jacobs disease, Huntington’s disease, prion-related diseases, and amyotrophic lateral sclerosis (ALS) [5,6].

The accumulation of misfolded proteins and proteins irreversibly modified by post-translational modifications (e.g., protein glycation, methionine oxidation, deamination of asparaginyl and glutaminyl residues) that induce conformational changes and impaired protein functions has also been observed in ageing [5]. Such alterations cannot be simply reversed by molecular chaperones, these may only accompany their substrates and by a stable association with their hydrophobic surfaces prevent their aggregation thus shifting the normal functions of HSPs from protein maintenance to fighting the growing number of damaged or non-functional enzymes [5,7]. The vulnerability of proteins to aggregation poses a great danger and the only solution in protecting the cell itself is their complete removal. However, ageing also decreases the activity of the proteasome, the main cytosolic proteolytic enzyme [8,9]. Therefore, some species developed constitutively induced chaperones (small heat shock proteins (sHSPs) and HSC70), which could fill the roles of impaired HSPs in ageing cells [10]. Under normal conditions, cytosolic HSC70 shuttles between the cytoplasm and the nucleus [11], which secures an export of nuclear proteins targeted for degradation [12]. The shuttling is transiently inhibited upon stress, when HSC70 becomes sequestered within the nucleus [13], thus protecting the stressed cells against possible damage and ensuring their survival when conditions improve [14]. When cells are exposed to excessive stress, a common response is to undergo cell death by either necrosis or apoptosis. HSPs are responsible for inhibiting both apoptotic and necrotic pathways and thus, they are also involved in cancer progression. The action of HSPs may cause uncontrolled cell growth, reduced tumor suppression, enhanced cell survival and may fuel tumor cell invasion, metastasis, and angiogenesis [15]. In various human cancers, HSPs were found to be expressed at high levels, providing an environment for tumor development and leading to poor patient prognosis and a resistance to therapy. Thus, HSPs could also serve as biomarkers of cancer formation in several tissues and show the degree of progression and aggression of several types of tumors [16].

Moreover, genetic mutations in genes encoding HSPs were shown as causative agents of several hereditary autosomal dominant or recessive disorders (e.g., hereditary spastic paraplegia SPG13, MitCHAP-60, EVEN-PLUS syndrome, and congenital sideroblastic anemia SIDBA4) suggesting that defects in mitochondrial chaperones exert a profound impact on overall mitochondrial biogenesis, integrity, and function.

## 2. General Characterization of Heat Shock Proteins

Heat shock proteins were first discovered in Drosophila almost 60 years ago [17]. They were initially recognized as proteins induced upon heat stress that were able to assist in the refolding of other proteins [18,19]. Further research soon expanded their roles to include aiding polypeptide folding, transfer across cellular and organellar membranes, assembly and disassembly of macromolecular complexes or aggregates and regulating their conformation and targeting proteins for proteolysis. Thus, from being simply heat stress factors, HSPs became molecular chaperones, though both terms are often still used as synonyms.

HSPs are found in all organisms, from prokaryotes to humans, and are found in eukaryotic compartments such as mitochondria, chloroplasts, and the endoplasmic reticulum [20]; they are often expressed constitutively even at physiological temperatures [21]. The HSP family was historically divided into sub-families based on molecular weight: HSP100, HSP90, HSP70, HSP60, HSP40 and small HSPs. The proteins of each sub-family share a similar domain structure, but each member associates with a unique set of substrates which have their own particular patterns of expression and cellular localization [17]. All molecular chaperones interact rather promiscuously with a broad range of unfolded proteins which are recognized by the abnormal exposure of hydrophobic amino-acid residues [2]. ATP hydrolysis is also essential for the activities of all HSP families except for sHSPs. These latter also seem to be predominantly expressed upon stress, while HSP70 and HSP90 come in both stress-induced and constitutively expressed forms [3]. Heat shock proteins are highly conserved with homologues present in E. coli, S. cerevisiae and humans. In fact, the chaperone system of eukaryotic mitochondria shares traits with its eubacterial ancestors.

Mitochondrial HSP60 and its co-chaperone HSP10, both belonging to the HSP60 sub-family, closely resemble the ring-shaped chaperonin GroE of *E. coli*, which is likewise composed of two proteins, GroEL and GroES [22]. Both are rather large, multimeric protein complexes, consisting of 14 subunits of HSP60 (GroEL) and a heptameric ring of HSP10 (GroES), which play a crucial role in the folding of nascent polypeptide chains [22,23]. Polypeptides of up to 60 kDa can be encapsulated within the central cavity of the HSP60 heptameric double-ring, which is covered by the co-chaperone HSP10. Subsequent conformational changes induced by ATP hydrolysis enable folding of the substrate without interference from the outer environment, which is particularly important for more hydrophobic proteins [24,25]. In addition to the typical stack of two heptamers, mitochondrial HSP60 can also function as a single ring [26] and is required for the proper folding of mitochondrial pre-proteins synthesized in the cytosol [27]. Here, HSP60 works in close cooperation with mitochondrial HSP70 (mtHSP70 or mortalin), a member of the HSP70 sub-family, which will be discussed later. HSP70 is primarily responsible for mediating the initial contact with newly synthesized polypeptides immediately after they cross the membrane or with stress-induced unfolded proteins in the mitochondrial matrix; if necessary, an interaction with HSP60 follows as a second step [2]. Mutations in *HSPD1*, the gene encoding human HSP60, cause severe neurodegenerative disorders, including an autosomal dominant hereditary spastic paraplegia, SPG13, characterized by late onset, progressive weakness and lower limb spasticity [28,29], and MitCHAP-60 disease, an autosomal recessive disorder with early onset, profound cerebral involvement, and lethality [30]. Moreover, HSP60 was shown to be involved in a variety of other pathologies, such as cardiovascular disease [31], chronic inflammation, autoimmune diseases, and cancer (review in [32]).

The HSP100 or ClpB sub-family (named after its *E. coli* representative), has a protective role against unwanted protein aggregation under diverse stresses [33]. The best studied mitochondrial homologue is the yeast HSP78 from *S. cerevisiae* (review in [34]). HSP78 is a representative of the AAA^+^ family (ATPases associated with diverse cellular activities) and has almost 50% similarity with *E. coli* ClpB. Interestingly, the enzyme has two nucleotide binding domains, NBD1 and NBD2, each with its own ATP-binding site, and a substrate binding domain (SBD) with a well-conserved Tyr251, which is essential for its function [34]. Like other AAA^+^ ATPases, HSP78 forms hexamers in the presence of ATP, although in purified mitochondria smaller oligomers were also found [35]. HSP78 is not itself essential for cell growth [36], but its deletion becomes lethal when combined with deletion or specific point-mutation of *HSP70* gene [37,38], which suggests some limited functional overlap. Further studies showed that HSP78, rather than protecting against aggregation during stress, is required for repairing mitochondrial functions after stress has ended [39]. Interestingly, higher eukaryotes, including mammals and humans, are for unknown reasons lacking an HSP78 homologue, with most of its functions being performed by HSP70 [40].

Representatives of mitochondrial proteases (review in [41,42]), which perform the final step of damaged protein removal, can be found in eubacteria, archaebacteria and eukaryotic organelles, such as the LON protease and the membrane-anchored metalloprotease FtsH (YTA10 in *S. cerevisiae* and AFG3L2 in humans). On the other hand, there are also proteases specific for particular organisms, e.g., the serine protease HTRA2 and the caseinolytic protease CLP are found in bacteria and higher eukaryotes, but not in most yeasts. In fact, only the CLPP protease subunit is missing in *S. cerevisiae* and *Schizosaccharomyces pombe*, the CLPX chaperone is still present [43,44]. The aerobic yeast *Yarrowia lipolytica* with an intact respiratory complex I possesses a functional CLPP, which suggests that CLPXP may have a role in regulating oxidative metabolism and mitochondrial respiration [45].

Usually, mitochondrial proteases form large ring-shaped homo- (e.g., YME1L1, LON) or hetero-oligomeric (e.g., AFG3L2 with the ATP-dependent zinc metalloprotease paraplegin, SPG7) complexes of 6–7 subunits with the active site located in the interior of the complex [41,46]. Proteolysis occurs when a substrate polypeptide translocates into the proteolytic chamber of the enzyme, mostly in an ATP-dependent manner. Since the opening of the ring-shaped complex is relatively small, the substrate can enter only in an unfolded state, which is ensured by either the chaperone properties of the protease itself (e.g., iAAA, mAAA, and LON protease) or the concomitant unfolding activity of a separate protein (CLPX in the CLPXP protease) [2,41].

Mitochondrial proteases are closely involved in modulating and regulating many essential mitochondrial functions [47], and their dysregulation is often associated with pathologies such as cancer, metabolic syndromes, or neurodegenerative disorders. In humans, at least 12 hereditary diseases are linked to a mutation in at least one of the genes encoding a known mitochondrial protease or pseudoprotease (review in [48]). For example, mutations in *AFG3L2* lead to spinocerebellar ataxia type 28 [49], or spastic ataxia 5 [50]; mutations in paraplegin are the causative agents of hereditary spastic paraplegia [51] and progressive external ophthalmoplegia [52]; and mutations in both AFG3L2 and SPG7 occur in dominant optic atrophy (DOA) [53]. In addition, early onset mitochondriopathy with development delay, muscle weakness, ataxia and optic nerve atrophy is caused by a homozygous missense mutation in *YME1L* [54] and a multisystemic developmental syndrome with cerebral, ocular, dental, auricular, and skeletal abnormalities (CODAS) arises from mutations in *LONP1* [55,56]. Intriguingly, mutations in *CLPP* and *CLPX* lead to rather different phenotypes. Whereas *CLPP* mutations cause Perrault syndrome, characterized by hearing loss, and ovarian failure, with ataxia, neuropathies, and intellectual disability in more severe cases [57,58], *CLPX* mutations promote erythropoietic protoporphyria [59].

Finally, the small HSPs are the most versatile group of chaperones, which show high variability both in sequence and size [60]. Small HSPs range from 12 to 43 kDa and are functionally able to interact with a number of substrates to prevent their aggregation under stress [3]. Nevertheless, they do share a conserved α-crystallin domain (named after the vertebrate eye-lens protein α-crystallin, the most prominent family member) in the C-terminal region of the protein [61,62]. In vivo, sHSPs form large oligomers of up to 24 subunits [63] and were shown to be constitutively expressed at very low levels, but strongly induced in response to heat stress.

In humans, three sHSPs (HSPB1/HSP27, HSPB4 and HSPB5) were shown to affect the CFTR (cystic fibrosis transmembrane conductance regulator) protein, whose mutations are the main cause of cystic fibrosis [64]. HSPB4 is also the major protein of the mammalian lens which, in cooperation with HSPB5, forms a dimer essential for lens transparency [65]; mutations in *HSPB4* lead to both peripheral and nuclear cataract diseases [66]. On the other hand, HSPB4 weakly expressed in pancreatic cells acts against tumorigenesis [67]. In mitochondria, HSPB1 (or HSP27) was shown to inhibit mitochondrial injury and apoptosis in normal and cancer cells. Moreover, HSPB1 emerges as a potent neuroprotectant in diverse neurological disorders, ranging from ALS to stroke. However, it is highly expressed in tumor tissues, which is often correlated with poor patient outcome (review in [68]). HSPB1, HSPB5 and HSPB8 are also highly expressed in mammalian cardiomyocytes [69], where they play at least a partial role in the cardiac stress response induced by ischemia/reperfusion injury and pressure-overload [70,71]. Interestingly, their phosphorylation seems to protect against the progression of heart failure. In the mitochondrial fraction of rat cardiomyocytes in the early stages of heart failure, HSPB1 (at Ser85) and HSPB5 (at Ser45 and Ser59) were found to be extensively phosphorylated by the serine protein kinases MAPKAPK2 (MAPK activated protein kinase 2) and ERK1/2 (extracellular-signal regulated kinase 1/2). In failing hearts, sHSP phosphorylation rapidly decreases, even though the amount of active protein kinases remains even higher than in normal hearts [72].

The most ubiquitous as well as the most abundant HSP, with representatives in all domains of life, are the HSP70 chaperones [73]. In eukaryotes, HSP70s are involved in a wide variety of cellular processes, mainly in folding of newly synthetized proteins, refolding of misfolded once, in protein transport into the organelles, and controlling of several regulatory enzymes. All these aspects are more thoroughly discussed in the following sections.

## 3. The 70-kDa Heat Shock Proteins (HSP70s)

Of all molecular chaperones, the HSP70 sub-family occupies a central position in every cellular proteostatic activity, from protein folding to disaggregation and degradation [74,75]. Its representatives are found in archaebacteria, prokaryotes, and eukaryotes, including plants and mammals [76], and possess one of the highest levels of conservation of all organismal proteins, with around 40–60% identity between the prokaryotic and eukaryotic homologues [77,78].

HSP70s reside in the cytoplasm of prokaryotes and in all major eukaryotic cellular sub-compartments [73,79,80]. At least one HSP70-encoding gene is expressed in all prokaryotes and eukaryotes. For example, *E. coli* harbors three Hsp70 isoforms: in addition to DnaK, the most thoroughly studied one, which mainly controls the folding of newly synthesized cytosolic proteins, two other HSP70 proteins have been identified, HscA and HscC [81,82]. HscA works together with its co-chaperone HscB in a chaperone/co-chaperone pair similar to that of DnaK/DnaJ [83,84]. It is required for the maturation of iron-sulfur clusters (ISC) [85,86] and is induced by cold stress [87]. HscC is heat induced, possesses ATPase activity, and is likely a chaperone [88], but does not act against denatured proteins [89]. *S. cerevisiae* expresses in total 11 HSP70 paralogues: 4 semi-redundant cytosolic/nuclear forms (SSA1, SSA2, SSA3, SSA4), 3 ribosome-associated chaperones (SSB1, SSB2, SSZ1), 3 mitochondrial chaperones (SSC1, SSQ1, SSC3) and 1 form specific for the endoplasmic reticulum (KAR2) [76,81,90,91]. Humans express 13 HSP70 homologues in different cellular compartments, including the cytosol and nucleus (HSPA1A/B, HSPA1L, HSPA2, HSPA6, HSPA7, HSPA8, HSPA12A/B, HSPA13, HSPA14), the ER (HSPA5) and the mitochondria (HSPA9) [81]. Human HSP70s differ not only in cellular localization but also in activity and expression [91,92]. Despite the large number of human isoforms, HSPA8 (also called HSC70) is the major, non-inducible cytosolic HSP70. It is constitutively expressed and provides the essential housekeeping functions in cellular protein quality control. Recent findings have also emphasized its involvement in regulating lysosome activity in specialized autophagy pathway called chaperone-mediated autophagy (CMA) (review in [93]). The second most abundant cytosolic homologue is the stress-inducible form of HSP70, HSPA1A (also named HSP72), whose expression increases in response to the accumulation of damaged or misfolded proteins [76,91,94].

In bacteria, deletions of *dnaK* often lead to cell death regardless of growth conditions [95,96,97,98,99,100], though *E. coli* does at least partially tolerate its absence when grown at optimal or low temperatures [100,101]. On the other hand, yeast mutants lacking all four canonical HSP70s are not viable at all [76,92]. Interestingly, the overexpression of only one SSA isoform in a *ssa1-4*Δ strain is sufficient to ensure cell viability, suggesting that SSA1–4 are partially functionally redundant [76,102]. In humans, complete knockouts of constitutively expressed HSP70s as HSPA5 (in ER), HSPA8 (in cytosol, HSC70), or HSPA9 (in mitochondria) are lethal [79,103].

In general, HSP70s consist of an N-terminal nucleotide-binding domain (NBD) and a C-terminal substrate-binding domain (SBD) connected by a flexible and highly conserved hydrophobic linker, which is crucial for allosteric inter-domain communication (review in [81]). The NBD domain is formed of four subdomains (Ia, IIa, Ib and IIb), organized into two lobes separated by a deep cleft, where the ATPase catalytic site resides (Figure 1a). In *E. coli*, the Ia NBD subdomain functions as a key mediator of inter-domain allostery, representing a signal transduction between the binding sites for ATP and substrate [104]. The SBD domain is capable of binding extended polypeptides rich in aliphatic residues and is composed of two parts, a β-sandwich subdomain (SBDβ) with the binding site and an α-helical subdomain (SBDα) acting as its flexible lid [81,94,105,106,107]. Both domains, NBD and SBD, are linked by a flexible linker that transfers the structural re-arrangements caused by the ATP hydrolysis from NBD to the SBD, which enables folding of the client protein [81].

All HSP70 sub-family members share at least two of the four structural features of the HSP70 archetype. Eukaryotic cells are structurally more complex than bacteria and as already mentioned, they contain more HSP70 family members. Cytosolic and nuclear HSP70s are characterized by a conserved, charged, but structurally disordered G/P rich C-terminal region harboring an EEVD sequence motif. This is involved in interactions with HSP70-specific co-factors, which, in turn, guide an HSP70-bound substrate towards either folding or degradation [74,110,111,112]. Other HSP70 members found in various cellular organelles (ER, mitochondria, chloroplasts) also contain specific N-terminal targeting signals that determine their localization.

For HSP70, two conformational states have been described, denoted as open and closed [74,80], or domain-docked and domain-undocked, respectively [113] (Figure 1a,b). In the closed state, ADP is bound in the nucleotide pocket of the HSP70 NBD and the SBD forms a closed cavity, binding client substrates with high affinity. The hydrolysis of ATP promotes NBD conformational changes, which are transduced through the linker domain to the SBD. The structural rearrangements trigger the clamping down/closure of the SBDα onto an unfolded protein, preventing its dissociation and allowing its folding. The replacement of ADP with ATP gives rise to further conformational changes leading to the open state, when the flexible linker becomes ordered, and the α-helical lid of the SBD is held open by an interaction with the ATPase region of the NBD [80,113].

The HSP70s function primarily as monomers, but they are tightly regulated by two types of co-chaperones [2,114]. Both are classified according to their bacterial homologues as (i) DnaJ-like or J-domain proteins (JDP) and (ii) GrpE-like nucleotide exchange factors, which appear in the endosymbiotic organelles. HSP70s were shown to require a bidirectional mechanism of nucleotide-dependent allosteric regulation, where ATP binding induces the release of the bound peptide, and substrate binding stimulates ATP hydrolysis [81,94,110] (Figure 2). HSP70 itself exhibits rather weak intrinsic ATPase activity, which increases in the presence of a protein substrate. However, the efficiency of the ATPase cycle is incomparably higher in the presence of its co-chaperones. The synergistic effect of a substrate binding with JDPs can stimulate the low basal ATP hydrolysis by more than 1000-fold [115], leading to a tighter binding of the substrate proteins to the HSP70 peptide-binding pocket. Additionally, the nucleotide-exchange factor promotes the release of imported precursors, and thus controls the overall cycle rate [81,114,116].

## 4. Human Mitochondrial HSP70 (mtHSP70)

Mitochondrial HSP70 (mtHSP70/GRP75/HSPA9/PBP74), also called mortalin, is an essential protein belonging to the HSP70 sub-family which has great importance for mitochondrial biogenesis and the correct functioning of the whole cellular machinery [17].

Mortalin was first identified in cell fusion studies of normal and immortal mouse fibroblasts as a marker of the mortal or lethal phenotype [117]. Mortalin was classified as a HSP70 stress chaperone based on its high degree of homology to other HSP70 members, including *E. coli* DnaK (51%), *S. cerevisiae* SSC1 (65%), and the rat cytosolic HSP70, HSC70 (46%) [117]. Two mortalin isoforms are found in mice, MOT-1 and MOT-2 [117], while humans have only one orthologue, which resembles MOT-2 [117]. Surprisingly, the two isoforms differ in only two amino acids in the C-terminal SBD (Val618Met and Arg624Gly) [118,119]. They also exhibit different cellular localizations and have different interactions with the tumor suppressor protein p53. In normal murine cells, MOT-1 was detected predominantly in the cytosol with no p53 co-localization, but in mouse immortal cell lines, MOT-2 was found in the perinuclear region near the nuclear membrane interacted functionally with p53, rendering the cells immortal [120,121,122].

In humans, mortalin is a 74 kDa (679 amino acid), constitutively expressed protein. It has often been described as “autophosphorylated” in a Ca^2+^-dependent manner [123,124,125], but the apparent phosphorylation seems to be the trapping of a radioactively labeled ATP in the HSP70 active site by the presence of calcium ions rather than the actual attachment of a phosphate group [126]. Regardless, mortalin is one of the most abundant proteins in the mitochondrial matrix, accounting for approximately 1% of its total protein content [127]. Although mortalin is predominantly found in the mitochondrial matrix, when overexpressed it can also be found in extramitochondrial sites, including the cytosol and the perinuclear region. Like its mouse homologue, mortalin in the perinuclear area was found to co-localize with the tumor suppressor protein p53 in several cancer cell lines, thereby sequestering p53 and inhibiting the expression of several important p53 target genes, such as CDKN1A (which encodes p21, cyclin-dependent kinase inhibitor 1), MDM2 (an E3 ubiquitin-protein ligase), BAX (an apoptosis regulator), or FAS (tumor necrosis factor receptor superfamily member 6) [122,124,128,129]. The increased cytosolic localization also enhances malignancy and tumor transformation [130].

Mortalin has the canonical HSP70 family structure, with a ~42 kDa NBD and a ~25 kDa PBD connected by a short hydrophobic linker (D^434^VLLLDVTP^442^), used for allosteric regulation by its co-chaperones [110,114,131]. Since mortalin is predominantly a mitochondrial protein, it also has a 46 residue mitochondrial pre-sequence at its N-terminus [132,133,134], but its C-terminus has the sequence K^671^EDQKEEKQ^679^, which differs from the EEVD motif found in other human HSP70 homologues [135].

The diversity of mortalin’s activities is shown by its large number of binding partners and its cellular localization. Mortalin’s functions are primarily regulated by interactions with its three co-chaperones: (i) HEP1, which prevents its self-aggregation and controls its ATPase activity [136], (ii) TID-1L and TID-1S isoforms, the J-proteins, which mediate substrate binding and synergistically trigger ATP hydrolysis [81], and (iii) the GRPE nucleotide exchange factor orthologues GRPEL1 and GRPEL2 [137] (Figure 2).

In mitochondria, mortalin performs two specific roles: as a chaperone and stress-survival factor, it assists in protein quality control by (re)folding or degrading non-functional proteins [138,139,140], and as an essential component of the presequence translocase-associated motor (PAM), it binds precursor proteins to promote their unidirectional [116,138,141,142] translocation across the two mitochondrial membranes and into the mitochondrial matrix [142]. An excellent review by Pfanner et al. [143] provides a more detailed characterization of the biogenesis of mitochondrial proteins. In addition to mortalin, the PAM complex contains TOM20 and TOM22, the receptors on the surface of the mitochondrial outer membrane which recognize the cytosolic precursors carrying the mitochondrial localization signal (these form positively-charged amphipathic α-helices) [144,145]. These cytosolic pre-proteins are subsequently transported through the major outer membrane protein translocation channel, TOM40, and are then engaged by the presequence translocase of the inner membrane, TIM23 [146,147,148]. In the PAM complex, mortalin is the central ATP-driven chaperone and, together with its co-chaperones, ensures the translocation of the pre-protein into the mitochondrial matrix [143]. In humans, mortalin was shown to directly interact with the TIM23 complex, as well as cardiolipin-containing lipid bilayers, enabling its insertion into the inner mitochondrial membrane [149]. Although mortalin is approximately ten times more abundant than TIM23 [143], only a small amount of it acts in the TIM23-associated PAM in driving pre-protein import; the majority is involved in mitochondrial protein folding [148]. The overall maintenance of mitochondrial homeostasis is ensured by the cooperation of mortalin with the HSP60–HSP10 chaperonin complex; together, these play a central role in the correct folding of matrix-localized proteins, preventing protein misfolding and promoting the refolding and proper assembly of unfolded polypeptides that are generated under stress conditions inside the mitochondria [150].

In addition, mortalin also plays an important role in iron-sulfur cluster (ISC) biogenesis within the matrix and the proper insertion of Fe-S apoproteins [151,152,153]. It also closely cooperates with other mitochondrial homeostasis maintenance factors, including the tumor necrosis factor receptor-associated protein type 1 (TRAP-1) [154,155] and the mitochondrial voltage-dependent anion channel (VDAC) [156]. It regulates mitochondrial properties, including ATP levels, membrane potential and permeability [157,158], and it associates with the mitochondrial contact site and cristae organizing system (MICOS) complex [159]. Moreover, together with HSP60 and the mitochondrial ATP-dependent protease LON, mortalin forms part of the peripheral region of human mitochondrial nucleoids [160].

Outside the mitochondria, mortalin is also involved in other cellular activities, such as regulation of p53 activity, calcium and ROS signaling, intracellular trafficking, control of centrosome duplication, differentiation, and many others [129,161,162,163,164,165].

## 5. Human mtHSP70 Co-Chaperones

### 5.1. HEP1

The mortalin escort protein HEP1 (also called Zim17/TIM15/DNLZ) is a small (15 kDa) DNL-type zinc finger protein that functions as a mtHSP70 partner on the matrix side of the inner mitochondrial membrane (Figure 2). HEP1 is frequently found in the mitochondria of eukaryotes, although no bacterial homologue is presently known [166,167]. Structurally, HEP1 forms asymmetric monomers, but it can also oligomerize depending on its actual concentration with Trp115 playing an important role in the process. Trp115, part of the zinc-biding domain (ZDB), lies on the surface of the monomer, but oligomerization buries it within the molecule between two α-helices at the C-terminal end of the protein; this sequesters the whole region from the outer environment when needed. The zinc finger motif seems particularly important for HEP1 itself, as well as for its interaction with mtHSP70. It has been shown that HEP1 is highly unstable in the presence of the EDTA chelator, indicating that the zinc ions are crucial for stabilizing its structure [136]. Moreover, when deletions or mutations in this motif occurs, HEP1 loses the ability to bind mortalin [168].

Studies in yeast demonstrated that HEP1 is essential for the mitochondrial import machinery. Deletion of its gene impaired the TIM23-dependent import of mitochondrial pre-proteins, which are necessary for yeast growth at elevated temperatures. When complexed with mtHSP70, HEP1 has a dual character: it protects mtHSP70 from self-aggregation, as mentioned above, and also participates in controlling its ATPase activity [167,169]. Zhai et al. [170] showed that His107, conserved in all mitochondrial and chloroplast HSP70-escort proteins, is especially required for ATPase stimulation. In humans, HEP1 binds the mortalin NBD directly [169], while in yeast, an interdomain linker is needed [167]. As noted above, the interdomain linker is a short loop of hydrophobic amino acids that connect the mtHSP70 NBD to its SBD, providing for their mutual communication [171]. Interestingly, the presence of human HEP1 increases the mtHSP70 ATPase activity by up to 49-fold [172], similar to the effect reported for the J-domain co-chaperones [173,174], and it also enhances the rate of nucleotide exchange, similar to the GRPE-type co-chaperones [175].

### 5.2. GRPE

In complex with mtHSP70, GRPE acts as a nucleotide exchange factor (NEF) (Figure 2), mediating the opening of the HSP70 nucleotide binding cleft to facilitate the dissociation of ADP, which, in turn, allows the binding of another ATP molecule and promoting the release of the protein substrate [137,176]. Generally, binding of GRPE to HSP70 can reduce its affinity for ADP by up to 200-fold [176,177].

In *E. coli*, GrpE acts as a NEF for the bacterial Hsp70 homologue DnaK [175]. In *S. cerevisiae*, MGE1 acts as a NEF for the yeast mtHSP70 SSC1 [178]. Mammals have two mitochondrial GRPE homologues, GRPEL1 (a 23 kDa protein) and GRPEL2 (25 kDa) [179] together with an additional exchange factor, BAG1, for cytosolic HSP70 [180]. The actual reason for such duplicity in mitochondria is still unknown, though several experiments have shown that GRPEL1 is essential for cell viability while GRPEL2 is not [181,182,183]. Konovalova et al. [183] found that GRPEL1 serves as the major exchange factor for mtHSP70 (mortalin), which is necessary for maintaining the proper translocation of proteins to the mitochondrial matrix through the PAM complex, while GRPEL2 acts as a “helper” protein. Konovalova et al. [183] also found that human GRPEL2 is a redox-sensitive protein, which can form dimers (through disulfide linkages) under oxidative stress. This suggests that even though GRPEL2 is not essential in cultured cells, it has may have adapted to fine-tune protein import and folding in response to altered redox conditions.

Human GRPE isoforms are approximately 43% identical and 65% similar to each other and share only moderate homology to bacterial GrpEs (25% identity and 45% similarity) [131]. To date, there is no crystal structure of human GRPE, though structural studies on the *E. coli* homologue GrpE showed that it forms an asymmetric homodimer when bound to DnaK, with one distal and one proximal monomer. The asymmetry arises from the proximal monomer, which is tilted towards the ATPase domain of DnaK. Its C-terminal domain forms a β-sheet and binds to the NBD of DnaK, thereby ensuring the release of ADP [137,184]. The function of the N-terminal 68 residues of GrpE is still elusive, though it is thought that this part might interact with the DnaK SBD and affect its dynamics. Thus, by interacting with both the DnaK NBD and SBD, GrpE stabilizes the DnaK–substrate complex, while simultaneously participating in ATP/ADP exchange [185,186,187]. Finally, residues 40–88 of GrpE form two α-helices which are thought to serve as a temperature sensor for regulating DnaK’s activity upon heat shock [137,188].

### 5.3. TID-1

TID-1 (tumorous imaginal disc protein 1), also known as DnaJ homolog subfamily A member 3 (DnaJA3), is a small mitochondrial protein belonging to the HSP40 sub-family. TID-1 is a human homologue of the bacterial protein DnaJ and the tumor suppressor protein TID-56 present in *Drosophila* [189,190]. Its translation occurs in the cytosol, and mRNA splicing produces two isoforms: a 43 kDa long isoform called TID-1L, and a shorter 40 kDa isoform called TID-1S. These isoforms differ in the length of their C-terminal ends (33 residues for TID-1L and 6 for TID-1S) [191,192]. NMR studies have shown that TID-1 contains a highly conserved J-domain at its N-terminus, typical for all DnaJ proteins, which is extremely important for its binding to mtHSP70 [193,194,195]. The J-domain contains approximately 70 amino acids and forms 4 α-helices with a characteristic conserved region, the HPD motif, located between the second and third α-helices. This region is responsible for both its binding to and stimulation of the mtHSP70 ATPase domain [196]. Adjacent to the J-domain, TID-1 possesses a G/F-rich region and a cysteine-rich region characterized by four repeats of a conserved CXXCXGXG motif with two coordinated zinc ions [197,198].

TID-1L resides mainly in the cytosol, where it interacts with cytosolic HSP70 and participates in the induction of apoptosis and various cellular signaling pathways. TID-1S, on the other hand, localizes primarily to the mitochondrial matrix, where it is responsible for mtDNA stability and maintains the mitochondrial membrane potential, thus acting against apoptosis [192,199,200,201]. TID-1 is one of the major co-chaperones of mtHSP70 and is mainly involved in stimulating its ATPase activity (Figure 2). Indeed, when other HSP70 co-chaperones are missing, a TID-1–mtHSP70 complex can still bind unfolded substrates and prevent their aggregation [191].

TID-1 is involved in numerous cellular processes, including cell growth, proliferation, differentiation, ageing, and survival [202,203,204,205]. In mammals, it is also involved in movement and plays an important role in the development of embryos and skeletal muscles [206,207,208]. Loss of TID-1 in heart results in cardiomyopathies and a decrease in mtDNA levels because it aids in the proper folding of DNA polymerase γ [209]. The protein is also involved in regulating mitochondrial homeostasis. Deregulation of TID-1 impairs its interaction with the mitochondrial dynamin-1-like protein DNM1L, causing fragmentation of the mitochondrial network [203] and negatively influences the CR6-interacting factor 1 (CRIF1) involved in the proper localization of the OXPHOS subunits into the inner mitochondrial membrane [210]. Similarly, an imbalance between TID-1 and mtHSP70 may be responsible for the mitochondrial fragmentation seen in patients diagnosed with optic atrophy 1 [211]. A recent study by Patra et al. [212] linked ataxia and developmental delay in one patient to a homozygous mutation in the *TID1* gene (Arg151Thr), which produced a TID-1 variant that was less efficiently imported into the mitochondria and had severely impaired co-chaperone functions.

## 6. Mortalin, Mortalin Co-Chaperones, Post-Translational Modifications and Human Diseases

### 6.1. PTMs

In recent decades, progress in proteomic analyses has allowed for more accurate studies of individual post-translational modifications (PTMs) in various cellular proteins. Even the early studies on molecular chaperones identified several PTMs, mostly phosphorylations, though their actual effects remained elusive [213,214]. PTM generally involves the addition of a small functional group (a phosphate, methyl, or acetyl group) to a specific side chain of a target protein, and it often occurs in response to changes in the cellular environment. By modifying proteins in this way, cells can react quickly to changes in environmental conditions without the need for the rapid and costly synthesis of new enzymes. On the other hand, deregulation of PTMs causes many human pathologies.

It was initially thought that the PTM of mitochondrial proteins was highly unlikely [215]. Later, it was found that mitochondrial proteins have many and many different PTMs, and that several of these modifications (e.g., phosphorylation, oxidation, nitrosylation, acetylation and other acylations) alter mitochondrial function. Typically, PTMs result from intra- and inter-cellular signaling and regulate the chemical network within the cell. In mitochondria, they have been shown to be involved in energy production, apoptosis, metabolism, and tissue response to ischemic injury [216]. The enzymes that modify mitochondrial proteins include the kinases and phosphatases responsible for phosphorylation and dephosphorylation events (review in [217]), an *O*-GlcNAc transferase responsible for *O*-GlcNAcylation, and SIRT5 (sirtuin-5), which acts in deacylations (desuccinylation, deglutarylation); some other occur non-enzymatically [216]. In mouse mitochondria, *S*-nitrosylation by the endothelial nitric oxide synthase (eNOS) found to be a common modification of proteins in the heart, brain, kidney, liver, lung, and thymus, which suggests that *S*-nitrosocystein-containing proteins are heavily used in metabolically active tissues [218]. More than 100 PTMs are currently known for the proteins of the human HSP70 sub-family, including the ubiquitous phosphorylation and acetylation, methylation, ubiquitination, SUMOylation, adenylation, ADP-ribosylation and others (review in [76]).

#### 6.1.1. PTMs of Mortalin

Returning to mortalin (mtHSP70), several PTMs of this protein were found in vivo, including phosphorylation, oxidation, and ubiquitination [161]. To take one example, treatment of rat hepatoma cells with aqueous peroxovanadate (pV), an inhibitor of tyrosine phosphatases which mimics oxidative stress, significantly enhances the selective tyrosine phosphorylation of mtHSP70, while pV treatment produced no such phosphorylation in the HSP70 sub-family ER chaperones GRP72 and BIP/GRP78 or the cytosolic HSC70 [219].

Further proteomic analyses revealed a plethora of other PTMs in mortalin (Figure 3A). The most abundant were acylations (acetylation, succinylation and mono-methylation) which were found on several lysines and arginines throughout the protein’s amino-acid sequence (Table 1 and Appendix A). Mortalin was also found to be phosphorylated at more than 50 serine, threonine, and tyrosine residues, five of which were identified in its mitochondrial targeting sequence (Ser3, Ser5, Thr22, Ser29 and Tyr46) [220]. The most frequently modified residue was Lys288, which was found to be acetylated in almost 90 mass spectrometry (MS) studies, predominantly in different types of cancer tissues (colorectal cancer, colorectal carcinoma, leukemia, chronic myelogenous leukemia, lung cancer and non-small cell lung adenocarcinoma [221]) and the heart rhythm disorder, ventricular tachycardia [220]. To date, however, no further experimental data are available on how these PTMs affect mortalin function.

Only two of the identified mortalin phosphorylation sites, Thr62 and Ser65 at the N-terminal end, are currently connected to a known kinase, a dual specificity protein kinase (TTK) called MPS1 (monopolar spindle 1) [222]. MPS1 is a widely conserved autophosphorylating protein kinase that is required for proper mitotic spindle assembly, checkpoint signaling, and several other functions during cell growth and differentiation [223]. Kanai et al. [222] showed that in humans, mortalin is phosphorylated at Thr62 and Ser65 in vivo and in vitro by *h*MPS1. The phosphorylated enzyme then super-activates *h*MPS1 through a positive feedback mechanism by directly bind it. This accelerates centrosome duplication.

A direct interaction between mortalin and fibroblast growth factor 1 (FGF1) was found in mouse embryonic fibroblasts, which was correlated with the level of its tyrosine-phosphorylation in late G1 phase. Phosphorylated mortalin increases its interaction with FGF1 by more than 4-fold and thus, might be the determining factor regulating the degree of mortalin–FGF1 complex formation in FGF1-induced cell growth and differentiation [224].

**Figure 3 ijms-22-08077-f003:**
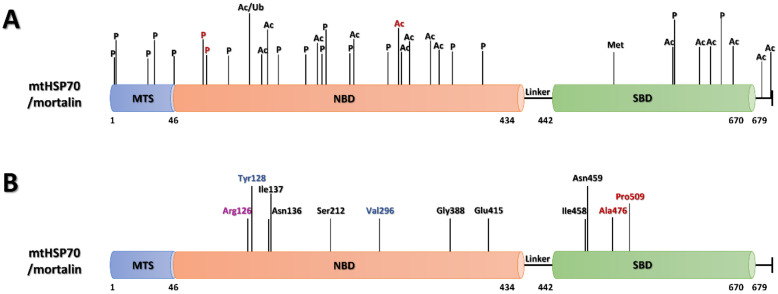
The structural domains of human mortalin highlighting currently known post-translational modifications (**A**) and modified amino-acid residues identified in human diseases (**B**). (**A**) The residue (acetylated Lys288) most frequently identified in MS searches for post-translational modifications [220], and Thr62 and Ser65 phosphorylated by a known protein kinase, MPS1 [222], are colored red. Only modifications identified by at least five separate proteomic studies are shown [220]. (**B**) Those amino acids whose mutations are involved in Parkinson’s disease [158,225], are shown in red; those residues whose modification is associated with EVEN-PLUS syndrome [226], are shown in blue. Arg126 (purple) was found to be modified in both, PD and EVEN-PLUS. The black amino acids were found to be modified in sideroblastic anemia, SIDBA4 [227]. The numbers below each diagram mark the beginning and end of the major divisions of the protein and help to indicate the position of a given amino acid in the protein sequence.

**Table 1 ijms-22-08077-t001:** The overview of mortalin post-translational modifications and their involvement in human diseases, which appear in the PhosphositePlus^®^ Database. Only modifications identified by at least five separate proteomic studies are listed below. A more detailed information about mortalin’s PTMs is included in Appendix A.

**Modification**	**Position**	**Localization**	**Diseases**
**Phosphorylation**	S3	MTS	breast cancer [228]
	S5	MTS	breast cancer [228]
	T22	MTS	breast cancer [228]
	S29	MTS	breast cancer [229]
	Y46	MTS	lung cancer [230]
	T87	NBD	lung cancer [231]
	Y128	NBD	gastric cancer, leukemia, chronic myelogenous leukemia, lung cancer
	S148	NBD	breast cancer [229,232]
	S162	NBD	lung cancer [233]
	S164	NBD	breast cancer [229], lung cancer [233]
	S200	NBD	breast cancer [229]
	T271	NBD	lung cancer
	S378	NBD	erythroid leukemia [234], acute myeloblastic leukemia [234], lung cancer, B cell lymphoma [234], non-Hodgkin’s lymphoma [234], multiple myeloma [234]
	S408	NBD	breast cancer [229]
	Y568	SBD	cervical cancer [235], gastric cancer, leukemia, chronic myelogenous leukemia, liver cancer, hepatocellular carcinoma, lung cancer
	S627	SBD	breast cancer [229], melanoma skin cancer [236]
**Mono-Methylation**	R513	SBD	gastric cancer, lung cancer
**Acetylation**	K121	NBD	lung cancer, B cell lymphoma
	K135	NBD	colorectal cancer, leukemia, chronic myelogenous leukemia, liver cancer, hepatocellular carcinoma, lung cancer [221]
	K138	NBD	leukemia, chronic myelogenous leukemia, lung cancer [221]
	K159	NBD	colorectal cancer, gastric cancer, leukemia, chronic myelogenous leukemia, lung cancer
	K206	NBD	not found
	K288	NBD	colorectal cancer, leukemia, chronic myelogenous leukemia, liver cancer, lung cancer [221], ventricular tachycardia
	K291	NBD	ventricular tachycardia
	K300	NBD	colorectal cancer, leukemia, chronic myelogenous leukemia, liver cancer, hepatocellular carcinoma, lung cancer [221]
	K345	NBD	gastric cancer, lung cancer [221]
	K567	SBD	colorectal cancer, leukemia, chronic myelogenous leukemia, lung cancer
	K600	SBD	liver cancer, lung cancer [221]
	K612	SBD	colorectal cancer, leukemia, chronic myelogenous leukemia, lung cancer [221]
	K646	SBD	gastric cancer, lung cancer
	K675	SBD	leukemia, chronic myelogenous leukemia
	K678	SBD	kidney cancer, leukemia, chronic myelogenous leukemia
**Ubiquitylation**	K121	NBD	multiple myeloma

#### 6.1.2. PTMs of Mortalin Co-Chaperones

Not much is currently known about the PTMs of the human mtHSP70 co-chaperones; most of what is known comes from broad-ranged MS proteomic analyses of a variety of cancerous cell lines and tissues.

A total of 21 modification sites were found for the human TID-1, 18 of which appear in the PhosphositePlus^®^ Database with three additional arginine methylations (Arg58, Arg238 and Arg293) given in UniProt [237] (Figure 4). Twelve of the modifications occur in the highly conserved N-terminal J-domain or close by, including three phosphorylations (Tyr113, Tyr114, Tyr142), one acetylation (Lys134), two ubiquitinations (Lys150, Lys152) and one succinylation (Lys130); close by there are two phosphorylations (Ser159, Ser169) one acetylation (Lys91), and one ubiquitination (Lys175) [220] (Figure 4). A second PTM cluster occurs at the C-terminal end of the protein, where six phosphorylations were identified (Ser287, Ser398, Tyr399, Tyr401, Tyr405, and Thr417); the most frequently phosphorylated TID-1 residue in lung and breast cancers, Ser398, was also found here [220,229,232] (Figure 4).

In the human GRPEL1 isoform, 12 PTMs have been identified so far, including two phosphorylations (Ser47, Thr181), succinylations (Lys72 and Lys100), six acetylations (Lys66, Lys94, Lys157, Lys162, Lys169, Lys215), and two ubiquitinations (Lys64 and Lys169); these are fairly evenly distributed along the whole protein (Figure 4). The most frequent modification is acetylation of Lys94, which was found in pulmonary cell lines predominantly associated with different types of lung cancer [220]. Curiously, the second human GRPE isoform, GRPEL2, was shown to be modified at completely different locations than GRPEL1, with a predominance of ubiquitinations (Lys71, Lys137, Lys160 and Lys172) and only one identified acetylation (Lys142) [220].

Likewise, only five phosphorylation sites were identified in the mtHSP70 escort protein HEP1, three of which appeared in the mitochondrial targeting sequence (Thr4, Ser14, Ser49), with the other two (Ser51 and Ser171) occurring at opposite ends of the HEP1 polypeptide chain [82,220] (Figure 4). Phosphorylation of Ser51 was found by the MS analysis of Jurkat T-lymphocytes and no further experimental data are presently available; phosphorylation of Ser171 was observed in various breast and lung cancer tissues, including HER2 positive breast cancer, luminal breast cancer types A and B, triple negative breast cancer, lung cancer, and non-small cell lung adenocarcinoma [220,229,233].

### 6.2. Mortalin, Co-Chaperones and Human Diseases

#### 6.2.1. Mortalin in Neurodegenerative Diseases

Given the pleiotropic functions of mortalin in mitochondrial biogenesis and protein regulation, it is not surprising that alterations in its expression, mutations, or post-translational modifications have been connected to a variety of pathologies.

The knockdown of mortalin in *Caenorhabditis elegans* leads to a premature ageing phenotype [238], while its overexpression extends the lifespan of both the nematode and human fibroblasts [239,240]. On the other hand, mutations in mortalin altered mitochondrial morphology, impaired mitochondrial membrane potential and increased ROS levels in neuronal and non-neuronal human cell lines [158]. In human neurons, an excessive generation of both RNS (reactive nitrogen species) and ROS contributes to neuronal injury and subsequent death, which is also accompanied by the accumulation of misfolded proteins, a characteristic feature of a number of neurological diseases, including neurodegenerative disorders (AD and PD), gliomas, and brain ischemia [241,242].

Increased protein oxidation levels were also observed in the senile plaque-dense brain regions of patients diagnosed with Alzheimer’s disease [243]. The expression of several heat shock proteins (HSP60, HSP70 and HSP90) was downregulated [244], apparently as a result of a reduced amount of heat shock factor 1 (HSF1), a key transcription factor responsible for the expression of HSP genes, in the cerebellum of AD brains [245]. Of these, mortalin was shown to be differentially expressed and phosphorylated in the hippocampus of AD patients as a cellular defense mechanism against oxidative stress [246]. When overexpressed, mortalin reduced the cell damage and apoptosis caused by the amyloid plaque formation characteristic of AD pathology [247]. Indeed, a higher level of oxidized mortalin was found in the hippocampi of AD-model mice, which are also widely accepted as a model of ageing, and has been implicated in several age-related diseases [248].

Using quantitative proteomics, it has been shown that mortalin expression is significantly decreased in the brain mitochondria of PD patients [249] with lower mortalin levels correlating with the disease progression [250]. Further genetic screens revealed three separate DNA aberrations in the *HSPA9* (mortalin) gene: missense mutations of Arg126 to tryptophan (R126W) in the ATPase domain (NBD) and Pro509 to serine (P509S) and Ala476 to threonine (A476T) in the substrate-binding domain [158,225] (Figure 3B). A study on yeast carrying analogous PD-associated mortalin variants showed that R126W and P509S lost chaperone function and that cells harboring the ATPase mutant showed a severe growth defect [251].

#### 6.2.2. Mortalin in Cancer

The expression of mortalin is also increased in several types of tumors and tumor-cell lines, suggesting it has a role in the initiation or progression of cancer [130]. High levels of mortalin are found in human brain tumors [252], breast ductal carcinoma [253], liver cancer [254], thyroid carcinoma [255], and colorectal adenocarcinoma (Table 1 and Appendix A); its up-regulation correlates with poor patient prognosis [256]. Interestingly, in normal brain tissue, mortalin is predominantly confined to neurons and is nearly undetectable in astrocytes, but in malignant astrocytoma, its expression rises with increasing tumor grade [252,257].

The same *HSPA9* substitutions, A476T and P509S, in human neuroblastoma cells exhibited loss of mortalin function, resulting in severe mitochondrial fragmentation and a concomitant reduction of mitochondrial mass. Acute mortalin depletion was moreover sufficient to induce apoptotic cell death, while the overexpression of either PINK1 (PTEN-induced kinase 1) or Parkin (E3-ubiquitin ligase), PD-related enzymes acting in mitophagy, was able to rescue the mitochondrial network in *HSPA9* knockdown neural cells [258].

Mortalin has also been shown to co-localize with duplicated centrosomes in the late G1, S and G2 phases of the cell division cycle. Its overexpression abolishes suppression of p53-mediated centrosome duplication, leading to malignant growth [259]. Mortalin binds p53 in the cytoplasm, thereby disabling its translocation into the nucleus and thus inhibits transcriptional activation of p53 and its control of centrosome duplication in cancer cells [260]. P53 suppression has major implications in carcinogenesis, as it leaves the cell vulnerable to uncontrolled proliferation arising from its inability to arrest the cell division cycle [122].

#### 6.2.3. Mortalin in Autosomal Recessive Diseases

In 2015, recessive mutations in the mortalin-encoding HSPA9 gene were identified in three children suffering from a genetic disorder named EVEN-PLUS syndrome (epiphyseal, vertebral, ear, nose, plus associated findings) [226]. Two of them (siblings) shared the homozygous HSPA9 R126W mutation mentioned above in connection with Parkinson’s disease, while the third bore a Y128C heterozygous mutation with severe V296* truncation of the enzyme on the second allele (Figure 3B). Both mutated sites are adjacent to one another on the surface of the protein and reside in the NBD quite far from the ATP/ADP binding site [226], but their effects seem to significantly influence the overall function of mortalin in vivo.

Interestingly, the vertebral and epiphyseal changes that characterize the EVEN-PLUS syndrome also appear for CODAS (cerebral, ocular, dental, auricular and skeletal) syndrome, another autosomal recessive disorder, caused by the mutations in *LONP1*, a gene encoding the mitochondrial protease LON [261,262]. Since mutations in completely different genes produce similar phenotypes in the CODAS and EVEN-PLUS syndromes, it may be that a common pathogenetic mechanism exists that produces a family of novel mitochondrial chaperonopathies [261].

In the same year, Schmitz-Abe et al. [227] reported that various changes in HSPA9, including frameshifts, in-frame deletions, and missense and nonsense mutations, were the causative agents of a congenital sideroblastic anemia (CSA) called SIDBA4. CSAs are characterized by the abnormal accumulation of iron in the mitochondria of erythroid cells and are collectively caused by mutations in genes encoding proteins engaged in mitochondrial heme synthesis and iron-sulfur (Fe-S) cluster (ISC) biogenesis. Affected individuals always possess a severe mtHSP70 loss-of-function allele with either Ser212Pro, Glu415Lys, Gly388Ser substitutions, Ile137* truncation, or an Ile458–Asn459 deletion combined with a milder missense allele, not an allele with a potential null mutation, which could be lethal [227] (Figure 3B). In addition to mortalin, mutations in GLRX5 (glutaredoxin 5) and ABCB7 (ATP-binding cassette transporter B7) were also shown to cause non-syndromic and syndromic forms of CSA, respectively [263,264], along with recently found aberrations in mitochondrial HSCB (heat shock cognate B), which is the partner of mortalin in ISC formation [265].

## 7. Conclusions

Several severe human diseases arise from alterations in mitochondrial functions. To overcome the problems caused by such conditions, mitochondria have developed several levels of protein quality control to preserve the overall organellar homeostasis. On the cellular level, mitochondria can trigger apoptosis to preserve the integrity of the tissue or organism, while on the organellar level, a selective form of mitophagy ensures the disposal of non-functional parts. At the molecular level, PQC comprises chaperones and proteases, which cooperate in maintaining proper protein folding and removing damaged proteins that might otherwise form toxic aggregates.

The mitochondrial HSP70 chaperone system consisting of mortalin and its co-chaperones HEP1, TID-1, and GRPE, forms one of the key components of PQC in mitochondria. As an essential protein that participates in proliferation, functional maintenance, and cellular stress response, mortalin has been implicated in many human pathologies, including neurodegenerative disorders, autosomal recessive diseases, and carcinogenesis. Indeed, recent studies indicate that mortalin could serve as a prognostic and predictive marker of cancer invasiveness as its up-regulation has been detected in a variety of malignancies, such as brain tumors, hepatocellular carcinoma, colon carcinoma, breast cancer, and leukemia. In addition to overexpression, mortalin and its co-chaperones undergo several post-translational modifications that can also contribute to the deregulation of mitochondrial homeostasis and promote pathological processes. So far, most of the known modifications were characterized by wide-range proteomic studies, and further investigation of their influence will require more detailed attention in the future.

## Figures and Tables

**Figure 1 ijms-22-08077-f001:**
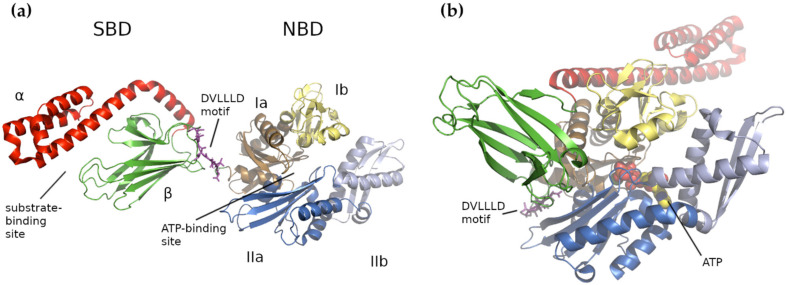
Structure of *E. coli* HSP70 DnaK. No complete structure of the human Hsp70 mortalin exists, so *E. coli* DnaK is used to illustrate the major structural features. (**a**) The Apo/ADP bound form with a compact substrate binding domain—closed state. The nucleotide-binding domain (NBD) is colored according to its four lobes: Ia is tan, Ib is pale yellow, IIa is marine blue and IIb is light blue. The α- and β-domains of the substrate-binding domain (SBD) are colored red and green, respectively. The substrate-binding and ATP-binding regions are indicated. The conserved DVLLLD linker is shown in magenta sticks. (**b**) The ATP-bound compact form showing an extended substrate-binding domain—open state. The bound ATP is shown as spheres and the DVLLLD motif is again indicated. Coloring is as in (a) and the orientation is roughly based on the location of the NBD in (**a**). (**a**) shows PDB structure 2KHO [108] and (**b**) is 4B9Q [109].

**Figure 2 ijms-22-08077-f002:**
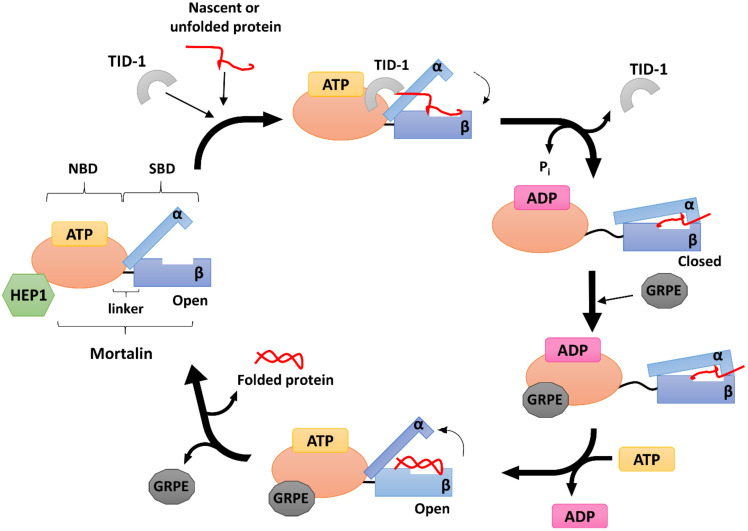
A likely model of the human mitochondrial HSP70 (mortalin) reaction cycle. Mortalin is composed of a nucleotide binding domain (NBD) and a substrate binding domain (SBD), which is divided into β and α sub-domains. The function of mortalin is tightly regulated by three co-chaperones, HEP1, TID-1 and either isoform of GRPEL1/2. HEP1 protects mortalin against self-aggregation and controls its ATPase activity. In the ATP-bound state, mortalin is in an open conformation and the flexible linker (located between the NBD and SBD) binds closer to the NBD. TID-1, which binds directly to mortalin, stimulates its ATPase activity, and prevents substrate protein aggregation. Substrate binding to the mortalin SBD binding cavity induces ATP hydrolysis and mortalin shifts into a closed conformation. GRPEL1/2 subsequently assists in the exchange of ADP for ATP. After ATP binding, mortalin returns into the open conformation and a folded substrate is released.

**Figure 4 ijms-22-08077-f004:**
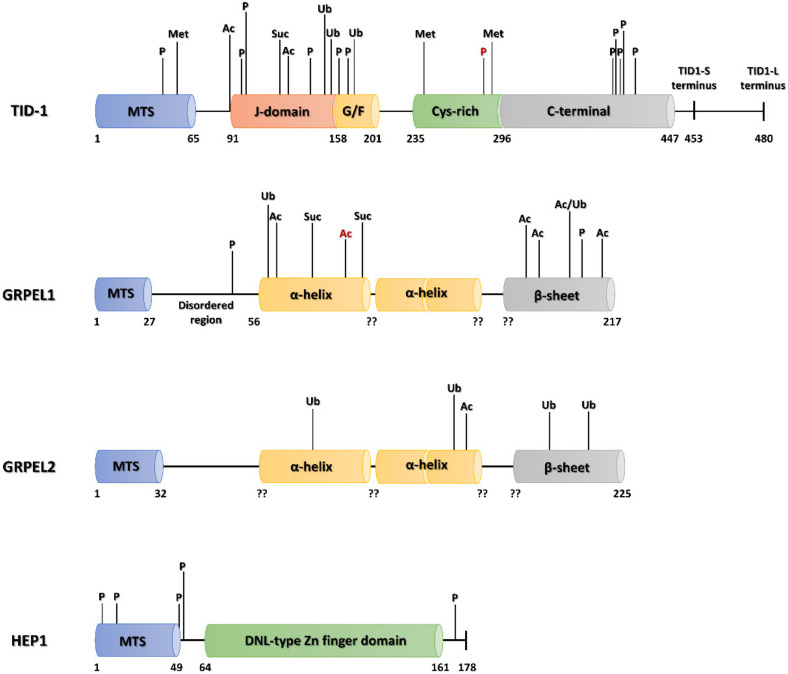
The structural domains of human TID-1, the GRPEL1/2 isoforms and HEP1 with currently known post-translational modifications highlighted [220]. The residues most frequently identified in MS searches for post-translational modifications are colored red. The number below each diagram mark the beginning and end of the major divisions of the protein and help to indicate the position of a given amino acid in the protein sequence. The question marks indicate that for the human homologue, the domain boundaries are not presently known.

## Data Availability

Not applicable.

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
