# Peer review of "Mitochondrial HSP70 Chaperone System—The Influence of Post-Translational Modifications and Involvement in Human Diseases"

_ijms, 2021, doi:10.3390/ijms22158077_

Round 1
Reviewer 1 Report
Ref: ijms-1301111
Title: Mitochondrial HSP70 Chaperone System - the Influence of Post-translational Modifications and Involvement in Human Diseases
Journal: IJMS-MDPI
The manuscript entitled: “Mitochondrial HSP70 Chaperone System - the Influence of Post-translational Modifications and Involvement in Human Diseases” by Havalová et al. is a well written article summarizing the roles of mitochondrial molecular chaperones with particular focus on the human mtHsp70 and its co-chaperones, PTMs and their involvement in human diseases. The article fits within the scope of the journal and would be useful for the readers working with chaperon systems and the impact in human diseases. Therefore, I recommend this manuscript to be accepted for publication in the IJMS journal, after some major revisions. Please find below the comments-suggestions that could be added to further improve the article below:
Specific major/minor comments:
Abstract: -Line 21: “impaired expression”: please replace to “deregulated expression”
Introduction:
-Line 32-33: “and the removal of proteins too damaged or misfolded to be repaired or refolded”: please rephrase since the meaning is not very clear in this sentence.
-Line 60-63: “The accumulation of misfolded proteins and proteins irreversibly modified by age-related post-translational modifications”: please explain more clearly how the PTMs are connected with ageing and the impact in age-related diseases.
-In the last paragraph of the introduction you should add a few lines mentioning the autosomal recessive diseases that you explain extensively later on.
- General characterization of heat shock proteins
-Lines 125-26: “autoimmune reactions, and cancer”: please correct to “autoimmune diseases and cancer”.
-Lines 207-208: “abundant HSP, with representatives in all domains of life, are the HSP70 chaperones [68]”: please replace the “domains of life” with other definition in the sentence. Also please include a few more lines to explain more why the focus of the manuscript is on HSP70 and not other HSPs.
-General comment: could be better to focus mainly in the mammalian-human heat shock proteins and expand more than in other types from other species (i.e yeast, bacterial etc)
- The 70-kDa heat shock proteins (HSP70s)
-I think this part could be more brief since the main focus of the review is the human HSP70s and the involvement in human diseases. Therefore, perhaps could be better to shorten this part and expand more on the PTMs later on.
- Human mitochondrial HSP70 (mtHSP70)
-A detailed figure showing the structural and functional domains described in this part for mortalin-mtHSP70 would help in the description of this part.
- Human mtHSP70 co-chaperones
-Please include the figure 1 in the text of this part as well since you included the molecular co-chaperons (TID-1, GRPE etc).
- Mortalin, post-translational modifications and disease
-Please rephrase the title as: “Mortalin, mortalin co-chaperones, post-translational modifications and human diseases”.
-Line 501-502: “…many different PTMs, and that several of these modifications alter mitochondrial function.”: please mention the most frequent PTMs here briefly in brackets.
-Line 513: “More than 100 PTMs are currently known…” please mention which of these have an impact in human diseases development; it would make sense to focus on those only.
-Figure 2A: should have been mentioned in the text. Figure legend should be more descriptive, mentioning which PTMs etc.
6.2. Mortalin in disease
-The title could be named as “Mortalin, co-chaperons and human diseases”.
-You should explain here clearly the effect of PTMs and other factors (mutations etc) of mortalin and co-chaperons in each disease separately.
-Also how about the combination of both PTMs for mortalin and co-chaperons; please elaborate whether the combined effect could be synergistic for the human disease development?
-Line 617, 638: the “gliomas” and “neuroblastoma” do not belong to the broad term of neurodegenerative diseases-please move this part in the section of cancer.
-Mortalin in cancer: it is important since this is the main focus of the manuscript to highlight here and expand on the effects of PTMs on mortalin and co-chaperons in cancer progression.
- . Mortalin in autosomal recessive diseases:-Lines 665-667: please move this part to the neurodegenerative diseases part since Parkinson`s belongs to this type of disease. Again in this part please include the effect of PTMs.
- Conclusion
-Table S1: this table is an important part and could be included as main table and not as a supplementary-perhaps by revising the table to be included as a summary table.
-Line 706: “hereditary diseases”: please replace with “autosomal recessive diseases”
-Line 710: “up-regulation of mortalin”: if upregulation is only overexpression please rephrase.
Author Response
We would like to thank the reviewer for a positive feedback on our manuscript and valuable comments and suggestions for further improvements. We have tried to address to all of his/her points as stated below. Please note that in the answers, a new numbering corresponding to the revised manuscript is used.

Reviewer 2 Report
This review describes the mitochondrial HDP70 chaperones, mortallin and its co-chaperones HEP1, TID-1, and GRPE.
Overall it provides useful information on the current status of mortallin and its role in proliferation, functional maintenance and cellular stress response. as well as recent indications that mortallin has been implicated in many human pathologies (neurodegenerative disorders, hereditary diseases, carcinogenesis) and could serve as a prognostic and predictive marker of cancer invasiveness.
Author Response
We would like to thank the reviewer for a positive feedback on our manuscript. We hope that the revisions based on the comments of other two reviewers will be acceptable.

Reviewer 3 Report
Havalová et al., have submitted the review entitled “Mitochondrial HSP70 Chaperone System – the Influence of Post-translational Modifications and Involvement in Human Diseases”.
The authors have made a great attempt to comprehend the mitochondrial heat-shock protein chaperone system.
The manuscript is very well written. Each section is well described and provided with enough background. The illustrations are very easy to follow and added value to the text.
Comments
Line 18-19: Expand the terms; HEP1, TID-1, and GRPEL1/2
Line 34-36: Please provide some examples of diseases in which this phenomenon occur; result of non-physiological temperatures or chemical modifications.
Line 37-39: Again please provide examples of diseases; accumulation of insoluble aggregates.
Line 67-69: This reference would fit here; https://doi.org/10.1038/s41598-018-34887-6
Line 234-236: This reference should be cited here; https://doi.org/10.3390/cells8080849
And should add a note about the importance of HSPA8.
Line 336-341: This information should be represented as an illustration.
The supplementary table is very informative and should be relocated to the main manuscript.
Author Response
Again, we would like to thank the reviewer for a positive evaluation of our manuscript and for all valuable comments and suggestions. We have addressed to all mentioned points as stated below. Please note that in the answers, a new numbering corresponding to the revised manuscript is used.

Round 2
Reviewer 1 Report
Ref: ijms-1301111-revised
Title: Mitochondrial HSP70 Chaperone System - the Influence of Post-translational Modifications and Involvement in Human Diseases
Journal: IJMS-MDPI
The manuscript entitled: “Mitochondrial HSP70 Chaperone System - the Influence of Post-translational Modifications and Involvement in Human Diseases” by Havalová et al. is a well written article summarizing the roles of mitochondrial molecular chaperones with particular focus on the human mtHsp70 and its co-chaperones, and their involvement in human diseases. The article fits within the scopus of the journal. The manuscript was improved after the revisions and the reviewers responded adequately to all the comments/suggestions. Therefore, I recommend this manuscript to be accepted for publication in the IJMS journal.